# Nonoverlap proportion and the representation of point-biserial variation

**Stanley Luck** [ID] *

Vector Analytics, LLC, Wilmington, DE, United States of America

* stan.luck@vectoranalytics.ai

**Data Availability Statement:** The data are available from FigShare, at https://doi.org/10.6084/m9.figshare.11591334.v2.

**Funding:** The author(s) received no specific funding for this work. The author, Stanley Luck, is a member of Vector Analytics, LLC, which is a

## Abstract

We consider the problem of constructing a complete set of parameters that account for all of the degrees of freedom for point-biserial variation. We devise an algorithm where sort as an intrinsic property of both numbers and labels, is used to generate the parameters. Algebraically, point-biserial variation is represented by a Cartesian product of statistical parameters for two sets of $\mathbb{R}^1$ data, and the difference between mean values ($\delta$) corresponds to the representation of variation in the center of mass coordinates, ($\delta, \mu$). The existence of alternative effect size measures is explained by the fact that mathematical considerations alone do not specify a preferred coordinate system for the representation of point-biserial variation. We develop a novel algorithm for estimating the nonoverlap proportion ($\rho_{pb}$) of two sets of $\mathbb{R}^1$ data. $\rho_{pb}$ is obtained by sorting the labeled $\mathbb{R}^1$ data and analyzing the induced order in the categorical data using a diagonally symmetric 2 × 2 contingency table. We examine the correspondence between $\rho_{pb}$ and point-biserial correlation ($r_{pb}$) for uniform and normal distributions. We identify the $\mathbb{R}^2$, $\mathbb{P}^1$, and $\mathbb{S}^1_+$ representations for Pearson product-moment correlation, Cohen's $d$, and $r_{pb}$. We compare the performance of $r_{pb}$ versus $\rho_{pb}$ and the sample size proportion corrected correlation ($r_{pbd}$), confirm that invariance with respect to the sample size proportion is important in the formulation of the effect size, and give an example where three parameters ($r_{pbd}, \mu, \rho_{pb}$) are needed to distinguish different forms of point-biserial variation in CART regression tree analysis. We discuss the importance of providing an assessment of cost-benefit trade-offs between relevant system parameters because 'substantive significance' is specified by mapping functional or engineering requirements into the effect size coordinates. Distributions and confidence intervals for the statistical parameters are obtained using Monte Carlo methods.

## 1 Introduction

This work began when we noticed that results from classification and regression tree (CART) analyses did not correspond well with statistical associations in genome-wide association studies (GWAS) [1]. Then, we discovered the extensive research literature discussing confounding properties of effect size measures used in our analyses. Statistical components of our bioinformatics system came from open source software packages that are widely used for research. In

science consulting company. Since there is no salary, Vector Analytics did not provide any funds for this work. Vector Analytics, LLC did not have any additional role in the study design, data collection and analysis, decision to publish, or preparation of the manuscript. The specific roles of the author are articulated in the 'author contributions' section.

**Competing interests:** The author, Stanley Luck, is a member of Vector Analytics, LLC, which is a science consulting company. This affiliation also does not alter our adherence to PLOS ONE policies on sharing data and materials. There are no competing interests connected with our consulting work at Vector Analytics, LLC. This work is not associated with any patents or commercial products.

data analysis, there are two important requirements for obtaining reproducible results. First, statistics methodology is subject to the general physical principle that it is necessary to account for all of the degrees of freedom when studying a quantitative phenomenon. Second, analysis protocols must correct for dependence on data acquisition parameters including unbalanced sample sizes, in order to obtain interpretable results for effect size. Our work on proportional variation and the phi coefficient for $2 \times 2$ contingency tables was recently published in this journal; we refer to this as Paper1 [2]. There, we demonstrate that odds-ratio or relative risk as standalone effect size measures, do not account for all of the degrees of freedom and are therefore subject to ambiguity. Using matrix factorization for the marginal sums, we identified the four alternative forms of proportional variation which serve as the basis for specifying the effect size. There is also an elementary discussion of projective geometry for fractional variation that might be helpful to the reader. Here, we study similar problems in the formulation of effect size for point-biserial variation and the associated correlation coefficient, $r_{pb}$. First, the term 'point-biserial' comes from psychology statistics, and we explain its use as a general reference for the two groups data analysis problem. The difference between mean values for two sets of $\mathbb{R}^1$ data, $\delta = \bar{y}_A - \bar{y}_B$, serves as the basis for specifying effect size for system response to perturbation. Statistically, analysis of $\delta$ corresponds to measuring the relation or association between a continuous variable and a binary categorical variable obtained by individually labeling the $\mathbb{R}^1$ data. The standard procedure is to replace the labels with numeric {0, 1} indicators. The Pearson product moment correlation coefficient ($r$) calculated from these numeric data is known as the point-biserial correlation coefficient ($r_{pb}$) [3]. This connection between $r_{pb}$ and $\delta$ explains our use of the term 'point-biserial'. It is standard terminology in the effect size literature. We provide a short discussion of the literature which gave us much inspiration, and note that there are several books on effect size methods as well [4, 5]. In their discussion of physical principles in the formulation effect size, Kelly & Preacher recommend that an effect size should serve as a sample size independent estimate of a system parameter [6]. The existence of alternative effect size measures, and their classification as relationship, group difference, and group overlap is discussed by Huberty [7]. A recently proposed group overlap measure is nonparametric but requires the use of kernel density estimators to produce an approximate representation of the unknown densities [8]. McGrath and Meyer give a nice review of research into the limitations of $r_{pb}$, and points out that different measures can "lead to different conclusions about the size or importance" of an effect [3]. Various researchers have already noted that there are two complications that can limit the range of $r_{pb}$. The first difficulty arises from the definition of $r_{pb}$, requiring the {0, 1} representation to allow the calculation of $r$. The {0, 1} representation corresponds to binary groupings of the data, comprising a pair of many-to-one mappings. The latter are incompatible with $r$ as a measure of the degree to which two variables are linearly related [9] and raises questions about the interpretation of $r_{pb}$. It has been shown that when the $\{y_A, y_B\}$ data are obtained by a dichotomy of a normal distribution, $r_{pb}$ has a maximum value of 0.79 [3, 10]. In contrast, when each $\mathbb{R}^1$ set corresponds to a normal distribution, $r_{pb}$ still ranges from $-1.0$ to $1.0$ [11, 12], with the proviso that the extremal values are reached in the limit as $|\delta|$ approaches infinity. Secondly, $r_{pb}$ is subject to confounding from unbalanced sample sizes for the $\{y_A, y_B\}$ data; in the effect size literature, the sample size proportions are usually referred to as 'base rates'. Then, variation in the sampling proportions between data sets leads to irreproducibility, which complicates the interpretation of $r_{pb}$. The machine learning community has rediscovered the problems associated with unbalanced sample sizes, creating the new term "classification imbalance" [13].

It is accepted practice to report a single effect size such as Cohen's $d$ as the basis for deciding the outcome of an experiment. However, $d$ is associated with an implicit parameterization that

does not account for all of the degrees of freedom for point-biserial variation, which results in ambiguity. Consequently, our objective is to construct a computational framework for a complete parameterization of the variation ($\mathbf{v}_{\text{pb}}$). We use an inductive approach based on connections between $r_{\text{pb}}$, Cohen's $d$, and the mean squared error information gain ($\text{IG}_{\text{MSE}}$). These measures play an important role because of their connections with elementary statistical concepts. We show that Cohen's $d$ is a perspective function of center of mass coordinates ($\delta, \mu$) for the mean value vector ($\bar{y}_{\text{A}}, \bar{y}_{\text{B}}$). We also identify a novel association measure, $\rho_{\text{pb}}$, which measures the degree of nonoverlap between two sets of $\mathbb{R}^1$ data.$\rho_{\text{pb}}$ is calculated directly from the data and is therefore nonparametric because the underlying densities are unspecified. A particular goal is to examine the dependence of $r_{\text{pb}}$ on unbalanced sample sizes because of concerns about the effect on reproducibility. We address other problems as well including the use of Monte Carlo methods to estimate the joint distribution for statistical parameters. As in Paper1, we use CART association graphs to compare the performance of various effect size measures. However, in this work we are particularly interested in the case where the target variable is a quantitative variable, which corresponds to the regression tree implementation (rCART) [14]. We show that $\rho_{\text{pb}}$ and the sample size proportion corrected correlation ($r_{\text{pbd}}$) serve as effect size measures for rCART while avoiding complications associated with $r_{\text{pb}}$. The main novel contributions of this work are as follows: 1) a computational model for generating statistical parameters for point-biserial variation $\mathbf{v}_{\text{pb}}$, which corresponds to the Cartesian product of parameters for two sets of $\mathbb{R}^1$ data, and identification of the fact that pure mathematics alone is not sufficient to specify a preferred effect size, 2) a sorting algorithm to estimate the nonoverlap proportion, $\rho_{\text{pb}}$, of two sets of $\mathbb{R}^1$ data using a diagonally symmetric $2 \times 2$ contingency table, 3) identification of the $\mathbb{R}^2$, $\mathbb{P}^1$, and $\mathbb{S}^1_+$ representations for Pearson correlation, 4) demonstration of the equivalence between $r_{\text{pb}}$ and $\text{IG}_{\text{MSE}}$, and 5) demonstration of the importance of adjusting for unbalanced sample sizes in impurity measures in rCART analysis.

## 2 Methods

The specification of a complete set of parameters for point-biserial variation, $\mathbf{v}_{\text{pb}}$, is a prerequisite for the rigorous formulation of effect size. Then, a measure for effect size is asociated with a perspective function of $\mathbf{v}_{\text{pb}}$. We begin with an examination of limitations of $r_{\text{pb}}$ in section 2.1. Then, we use an inductive approach to construct an algebraic framework for point-biserial variation in four sections 2.2–2.5.

### 2.1 The effect of unbalanced sample sizes on $r_{\text{pb}}$

The derivation and limitations of $r_{\text{pb}}$ are reviewed by McGrath and Meyer [3]. Two sets, $\mathbf{y}_{\text{A}} \in \mathbb{R}^{N_{\text{A}}}$ and $\mathbf{y}_{\text{B}} \in \mathbb{R}^{N_{\text{B}}}$, are combined to form a set of paired values, $\{(c_i, y_i) | c_i = 'A' \lor 'B', y_i \in \mathbb{R}^1, 1 \leq i \leq N, N = N_{\text{A}} + N_{\text{B}}\}$, where $c_i$ is a group membership label, and the $\{(c_i, y_i)\}$ data correspond to the vectors, $(\mathbf{c}, \mathbf{y})$. The standard practice is to invoke a numeric $\{0, 1\}$ representation for $\mathbf{c}$ to obtain an indicator vector, $\mathbf{I}_c \in \mathbb{R}^N$. Then, application of the Pearson product-moment formula produces the point-biserial correlation coefficient [3]

$$r_{\text{pb}} = \frac{\delta}{\sqrt{\delta^2 + \frac{S_p^2}{p_{\text{A}} p_{\text{B}}}}}, \tag{1}$$

$$= \frac{\sqrt{p_{\text{A}} p_{\text{B}}} d}{\sqrt{p_{\text{A}} p_{\text{B}} d^2 + 1}}, \tag{2}$$

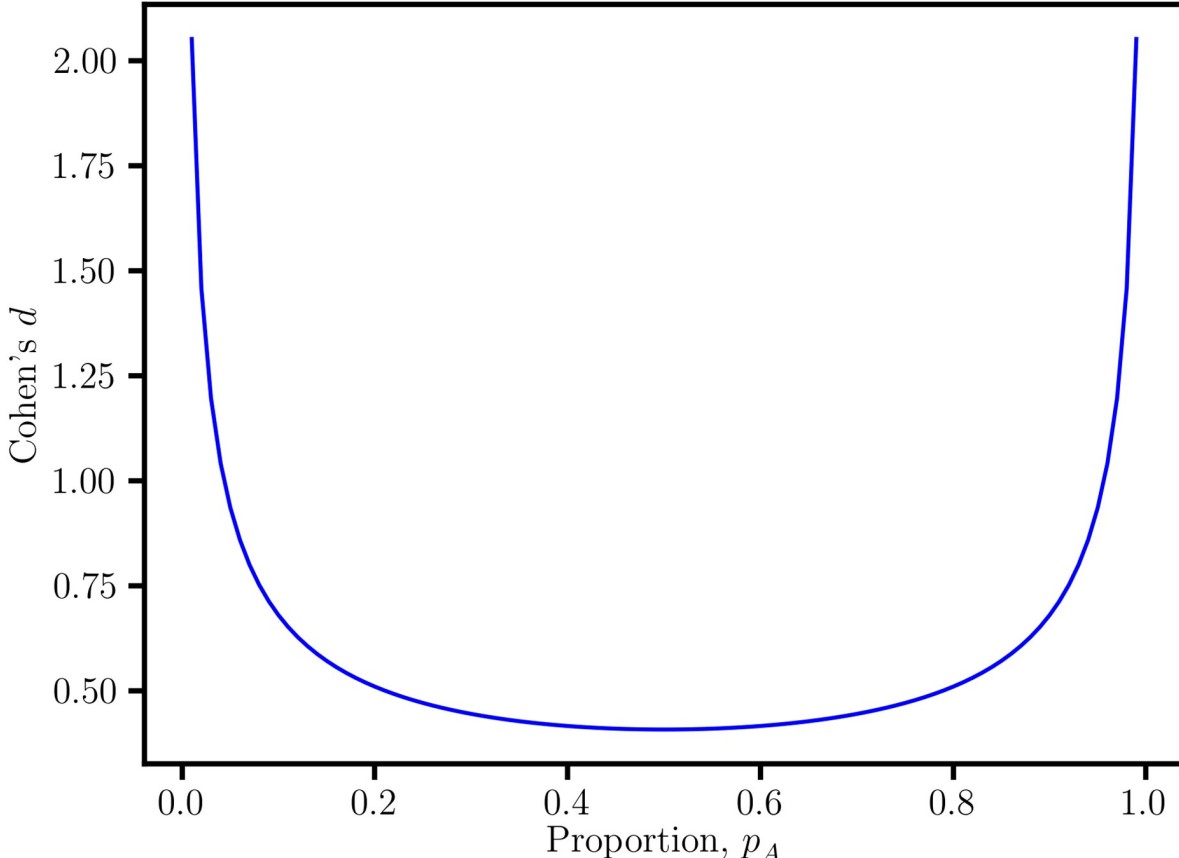

**Fig 1. Quadratic dependence of the point-biserial correlation coefficient, $r_{pb}$.** For the fixed value $r_{pb} = 0.2$, there is a range for Cohen's $d$ and the sample size proportion, $p_A$. This ambiguity complicates the interpretation of $r_{pb}$ as an effect size measure.

where $p_A = N_A/(N_A + N_B)$ and $p_B = 1 - p_A$ are sample size proportions, Cohen's $d$ is defined as

$$d = \frac{\delta}{S_p}, \tag{3}$$

and the pooled variance is the weighted average of the sample variances, $S_p^2 = p_A S_A^2 + p_B S_B^2$. Thus, $|r_{pb}|$ approaches unity as $|d| \to \infty$ [11, 12] for $0 < p_A < 1$. Rearranging Eq 2, we obtain the quadratic relation

$$d^2(1 - r_{pb}^2) - \frac{r_{pb}^2}{p_A p_B} = 0. \tag{4}$$

For a fixed value of $r_{pb}$, there is a range of $(d, p_A)$ values (Fig 1). Alternatively, the variation in $(r_{pb}, p_A)$ for fixed $d$ becomes a source of irreproducibility in $r_{pb}$ because $p_A$ can vary between experiments depending on the data acquisition protocol. This ambiguity explains why researchers have expressed concern about the confounding effect of unbalanced sample sizes on $r_{pb}$, and effect size in general [3, 6]. Furthermore, the binomial $p_A p_B$ dependence originates

from the covariance

$$\mathrm{Cov}(\mathbf{I_c}, \mathbf{y}) \quad = \overline{I_c y} - \overline{I_c}\,\bar{y}, \tag{5}$$

$$= p_A p_B (\bar{y}_A - \bar{y}_B), \tag{6}$$

and variance, $\mathrm{Var}(\mathbf{I_c}) = p_A p_B$. Therefore, the criticism about $p_A p_B$ dependence applies more broadly to the use of the numeric $\{0, 1\}$ indicator variable. Various researchers have already recommended that the proportions should be equalized, $p_A = p_B = 1/2$, in Eq 2 to give [3]

$$r_{\mathrm{pbd}} \quad = \frac{d}{\sqrt{d^2 + 4}}. \tag{7}$$

This 'attenuation-corrected' coefficient is denoted as $r_c$ in [4]. The $r_{\mathrm{pb}}$ and $r_{\mathrm{pbd}}$ curves in Fig 2 provide an illustration of this correction. The one-to-one projective relation between $r_{\mathrm{pbd}}$ and Cohen's $d$ is discussed in section 2.4, and the application of $r_{\mathrm{pbd}}$ in rCART is discussed in section 2.5.

## 2.2 Statistical parameters for point-biserial variation

In this section, we consider the question of how to generate a set of parameters for statistical variation in point-biserial data. The fact that $r_{\mathrm{pb}}$ is subject to confounding effects suggests that replacing categorical labels with $\{0, 1\}$ numeric values is an improper procedure, because the labels acquire arithmetic properties in an ad-hoc way. Instead, we propose a new framework where sort is used as an intrinsic property of both numbers and labels. Suppose there is a machine which generates numbers with labels, $(c_i, y_i)$, in no particular order, placing them in a data table to produce a point-biserial data set. Then, the table can be sorted using either $\mathbf{c}$ or $\mathbf{y}$, to obtain orderings denoted as $\mathbf{y_c}$ and $\mathbf{c_y}$, respectively. As we discuss next, these orderings are associated with statistical parameters, $\mathbf{v_c}$ and $\mathbf{v_y}$, respectively. However, there is no rule that

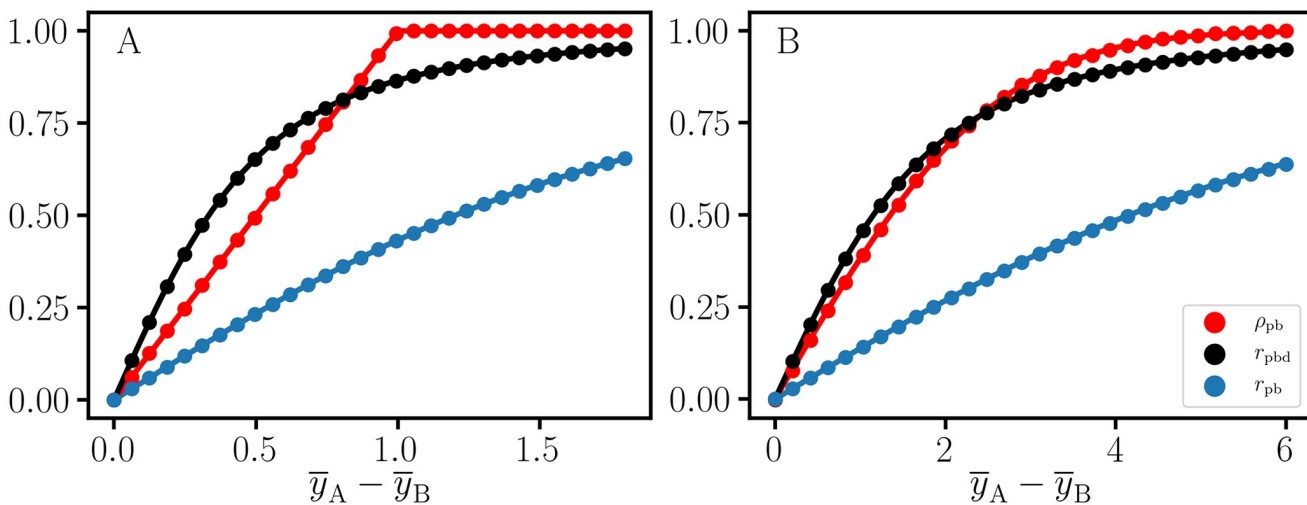

**Fig 2. Nonoverlap proportion and point-biserial correlation.** Theoretical curves and estimated values for point-biserial correlation, $r_{\mathrm{pb}}$, nonoverlap proportion, $\rho_{\mathrm{pb}}$, and sample size adjusted correlation, $r_{\mathrm{pbd}}$, for simulated data with unequal sample sizes ($N_A : N_B = 15000 : 500$) and the difference between mean values, $\bar{y}_A - \bar{y}_B$. Compared to $r_{\mathrm{pbd}}$, $r_{\mathrm{pb}}$ is attenuated due to the confounding effect of the binomial sampling factor. A: Uniform unit width ($\sigma = 1/\sqrt{12}$) distributions. B: Standard normal ($\sigma = 1$) distributions.

specifies which parameterization, $\mathbf{v_c}$ or $\mathbf{v_y}$, might be preferred. Therefore, we make the following proposition,

**Proposition 1**. *Point-biserial variation is parameterized by the Cartesian product of statistical parameters for the* $\mathbf{y_c}$ *and* $\mathbf{c_y}$ *orderings,*

$$\mathbf{v}_{\mathrm{pb}} = (\mathbf{v_c}, \mathbf{v_y}). \tag{8}$$

The $\mathbf{y_c}$ ordering corresponds to sorting the $\mathbf{y}$ data into two sets, $\mathbf{y_c} \leftrightarrow \{\mathbf{y_A}, \mathbf{y_B}\}$. Then, the statistical parameters for the two sets are associated with a two-component Cartesian product structure, yielding the familiar effect size measures, Cohen's $d$ and $r_{\mathrm{pb}}$ as discussed in section 2.3. The $\mathbf{c_y}$ ordering is associated with a new nonoverlap measure, $\rho_{\mathrm{pb}}$. The two types of $\mathbf{y}$-sort, ascending or descending, produce orderings where either $\{(c_i, y_i)|y_i \leq y_{i+1}\}$ or $\{(c_i, y_i)|y_i \geq y_{i+1}\}$, respectively. Then, the $\mathbf{c}$-column corresponds to a $\mathbf{y}$-ordered string, $\mathbf{c_y}$. The induced order from the $\mathbf{y}$-sorting is reflected in the degree of mixing of As and Bs in $\mathbf{c_y}$. Next, we sort the data with respect to $\mathbf{c}$ obtaining a maximally ordered string, $\mathbf{c_y}$, where the As and Bs are completely separated. $\mathbf{c}_M$ corresponds to the condition where $\mathbf{y_A}$ and $\mathbf{y_B}$ are disjoint in $\mathbb{R}^1$, which has been characterized as "perfect correlation" [11]. Our $\mathbf{c_y}$-sorting algorithm requires equal sample sizes, $N_A = N_B$. When the sample sizes are unequal, a preprocessing step is required. Suppose $N_B < N_A$. Then, the $\mathbf{y_B}$ data are replicated to create a new data set, $\mathbf{y}_{\mathrm{Brep}}$, such that $N_{\mathrm{Brep}} = N_A$. If the difference in sample size is small, $0 < N_B - N_A < N_B$, then a subset of $\mathbf{y_B}$ uniformly spaced by rank is replicated. The $\mathbf{y}_{\mathrm{Brep}}$ and $\mathbf{y_A}$ data are combined to obtain the $(\mathbf{c_y}, \mathbf{c}_M)$ strings. They constitute a set of joint observations for two categorical variables, which are summarized in a diagonally symmetric $2 \times 2$ contingency table of the form $[[a, b], [b, a]]$. The symmetric form results from the equal sample size condition, which requires that the rows and columns each sum to $N_A$. Then, the nonoverlap proportion is given by the difference in proportions

$$\rho_{\mathrm{pb}} = p_a - p_b, \tag{9}$$

where $p_a = \frac{a}{a+b}$, and $p_b = 1 - p_a$. When $\mathbf{y_A}$ and $\mathbf{y_B}$ are disjoint, $|\rho_{\mathrm{pb}}| = 1$. The sign of $\rho_{\mathrm{pb}}$ is arbitrary because the order of the columns (or rows) of the $2 \times 2$ table depends on the direction of the sort in $\mathbf{y}$ or $\mathbf{c_M}$. In our implementation, the sign is chosen to be consistent with Cohen's $d$. The $\rho_{\mathrm{pb}}$ values in Fig 2 were obtained using this sort algorithm. The overlap between uniform unit width $(\sigma = 1/\sqrt{12})$ distributions is an important pedagogical case because the expressions for Cohen's $d$, $r_{\mathrm{pbd}}$, and $\rho_{\mathrm{pb}}$ take a simple form. Geometrically, the overlap $(\theta_{\mathrm{U}})$ is given by a rectangle with area $\theta_{\mathrm{U}} = 1 - \delta$ for the difference between mean values, with $0 \leq \delta \leq 1$, and $\theta_{\mathrm{U}} = 0$ for $\delta > 1$. The nonoverlap is given by $\rho_{\mathrm{pbU}} = 1 - \theta_{\mathrm{U}} = \delta$, with $0 \leq \delta \leq 1$. Similarly,

$$d_{\mathrm{U}} = \sqrt{12}\delta, \tag{10}$$

$$r_{\mathrm{pbdU}} = \frac{\delta}{\sqrt{\delta^2 + 1/3}}. \tag{11}$$

For the overlap of standard normal $(\sigma = 1)$ distributions, we obtain

$$d_{\mathrm{N}} = \delta, \tag{12}$$

$$r_{\mathrm{pbdN}} = \frac{\delta}{\sqrt{\delta^2 + 4}}, \tag{13}$$

$$\rho_{\mathrm{pbN}} = 2\Phi(\delta/2) - 1, \tag{14}$$

where $\Phi$ is the cumulative normal distribution function [8]. In Fig 2, we observe that at a large enough $\delta$, $r_{\mathrm{pbd}}$ is attenuated compared to $\rho_{\mathrm{pb}}$, as expected [11]. However, for small $\delta$, the inequality is reversed, i.e., $r_{\mathrm{pbd}} > \rho_{\mathrm{pb}}$. Nevertheless, there is close correspondence between $r_{\mathrm{pbd}}$ and $\rho_{\mathrm{pb}}$ for both the uniform and normal distributions. This is particularly true for highly correlated data where both $r_{\mathrm{pbd}}$ and $\rho_{\mathrm{pb}}$ are near 1, and are therefore equivalent. However, in section 3 we demonstrate that when the data are not well correlated, both $r_{\mathrm{pbd}}$ and $\rho_{\mathrm{pb}}$ are needed in order to distinguish different forms of point-biserial variation. We conclude that $r_{\mathrm{pbd}}$ and Cohen's $d$ serve as measures of the nonoverlap of distributions but are not necessarily equivalent to $\rho_{\mathrm{pb}}$.

## 2.3 Coordinates for a two-component system of distributed effects

In this section, we discuss the fact that $d$ and $\rho_{\mathrm{pb}}$ are only two elements of a minimal set of parameters for representing point-biserial variation. The one-to-one correspondence, $d \leftrightarrow r_{\mathrm{pbd}}$, will be discussed in section 2.4. Algebraically, $\mathbf{v_c}$ corresponds to the Cartesian product of statistical parameters for two sets of $\mathbb{R}^1$ data, $\mathbf{v_c} = (\bar{y}_{\mathrm{A}}, S_{\mathrm{A}}^2, N_{\mathrm{A}}) \times (\bar{y}_{\mathrm{B}}, S_{\mathrm{B}}^2, N_{\mathrm{B}})$. Introducing the center of mass parameter, $\mu = (\bar{y}_{\mathrm{A}} + \bar{y}_{\mathrm{B}})/2$, the mean values vector is expressed as

$$(\bar{y}_{\mathrm{A}}, \bar{y}_{\mathrm{B}}) \quad = (\mu + \delta/2, \mu - \delta/2), \tag{15}$$

$$= \mu(1,1) + \frac{\delta}{2}(1,-1), \tag{16}$$

where $(1, 1)$ and $(1, -1)$ comprise the center of mass basis. We note that the generalization for a weighted average is straightforward. A similar decomposition holds for variances

$$\left(S_{\mathrm{A}}^2, S_{\mathrm{B}}^2\right) = S_{\mu}^2(1,1) + \frac{S_{\delta}^2}{2}(1,-1), \tag{17}$$

where $S_{\mu}^2 = (S_{\mathrm{A}}^2 + S_{\mathrm{B}}^2)/2$ and $S_{\delta}^2 = S_{\mathrm{A}}^2 - S_{\mathrm{B}}^2$. A further reduction is obtained if the variances are homoscedastic, $S_{\mathrm{A}}^2 = S_{\mathrm{B}}^2$, yielding $S_p^2 = S_{\mu}^2$, and $S_{\delta}^2 = 0$. Finally, we obtain

$$\mathbf{v}_{\mathrm{pb}} = (\delta, \mu, S_{\mu}^2, S_{\delta}^2, \rho_{\mathrm{pb}}, p_{\mathrm{A}}, N_{\mathrm{A}} + N_{\mathrm{B}}), \tag{18}$$

as a minimal set of parameters for point-biserial variation. However, we observe that $\mathbf{v}_{\mathrm{pb}}$ is not unique because functions of the components, $\{f_i(v_{\mathrm{pb},i})\}$, including linear fractional transformations can be introduced to obtain alternative representations. Mathematics alone is not sufficient to specify a preferred vector basis, which explains why there are alternative effect size measures [6, 7]. Furthermore, $r_{\mathrm{pb}}$ and Cohen's $d$ correspond to perspective functions [15] of $\mathbf{v}_{\mathrm{pb}}$ and do not account for all of the degrees-of-freedom. Consequently, the practice of using one of these measures to serve as a one-parameter summary of experimental results will be subject to irreproducibility.

The term 'substantive significance' has been used to refer to the magnitude of an effect that would be regarded as practically important in a given application [6]. Suppose functional or engineering requirements are expressed in terms of a vector, $\mathbf{h}$, of system parameters. Then, the utility of an effect would be specified as a mapping, $u : \mathbf{h} \mapsto \mathbb{R}^1$. The specification of $u(\mathbf{h})$ would account for differences in cost-benefit trade-offs for variation in the $\{h_i\}$ components. The substantive significance for the effect size would be determined by the mapping, $u(\mathbf{h}) \to u(\mathbf{v}_{\mathrm{pb}})$. Without this information, it is difficult to reach a consensus on the merits of an effect size. This explains the criticism of Cohen's thresholds for small, medium, and large effects as "somewhat arbitrary" [16] and suggests that the significance of the magnitude of an effect size depends on the research question [3, 17, 18].

A fundamental limitation arises from the fact that the $(\delta, \mu)$ center of mass decomposition does not extend to higher dimensions in a straightforward way. Consider the group means vector for three sets, i.e., $(\bar{y}_A, \bar{y}_B, \bar{y}_C)$. The default center of mass parameter is defined as $\mu = (\bar{y}_A + \bar{y}_B + \bar{y}_C)/3$. However, there is no standard procedure for choosing the two additional deviation parameters needed to specify a complete basis. Consequently, the formulation of an effect size measure for multiple group variation is not a well-posed problem, i.e., there is no unique solution [19]. This explains why Cohen's $d$ does not generalize to schemes involving more than two groups [20] and provides support for previous recommendations to break down 'complicated hypotheses', p. 526 [21], and 'reduce any multiple-level or multiple-variable relationship' into a set of two-variable effect size relationships [17]. This provides the raison d'être for the development of exploratory methodologies such as CART in high-dimensional data analytics [22, 23].

## 2.4 Homogeneous coordinates for Pearson correlation

In the effect size literature, it is accepted practice to distinguish three different types of effect size measure, 'relationship', 'group difference', and 'group overlap' [3, 7]. In this section, we discuss the fact that this classification is misleading. We have already discussed the fact that Cohen's $d$, $r_{pbd}$ and $\rho_{pb}$ all serve as measures of nonoverlap (section 2.2). Now, we point out that $r_{pbd}$ and Cohen's $d$ are two sides of the same coin because relationship and group difference correspond to different coordinate systems for representing fractional variation. Such correspondences are quite useful in exploring statistical dependence in high-dimensional data. Consider a vector $(a, b) \in \mathbb{R}^2$. Division by the y-component produces the ratio vector, $\{\alpha = (\alpha, 1) \in \mathbb{P}^1 | \alpha = a/b, b \neq 0\}$. Ratios can be distinguished by their representations as points in the projective line, $\mathbb{P}^1$. However, normalization of a ratio vector by the Euclidean length, $\|\alpha\| = \sqrt{\alpha^2 + 1}$, produces the unit vector $\hat{\alpha}$, which is a point in the positive half-circle $\mathbb{S}^1_+$. Thus, a fractional quantity can be represented as a point in either $\mathbb{P}^1$ or $\mathbb{S}^1_+$. Algebraically, the $\mathbb{P}^1$ and $\mathbb{S}^1_+$ representations are related by linear fractional transformations. In the terminology of projective geometry, a ratio corresponds to a perspective function, $P(\mathbf{u}, t) = \mathbf{u}/t$, for vector $\mathbf{u}$ [15]. The scaling invariance property of $\alpha$ is represented by the equivalence relation

$$\alpha \frac{b}{t} - \frac{a}{t} = 0,$$

with $t \neq 0$. Geometrically, this relation specifies points on the line passing through the origin, $(a, b)$ and $(\alpha, 1)$. The points, $(a, b)t$, constitute the homogeneous coordinates [24] for the line. The homogeneous coordinates concept shows that there is a natural correspondence between 'relationship' and 'group difference' effect size. Expressing the Pearson product-moment correlation coefficient as the rescaled covariance [9]

$$r = \frac{\text{Cov}(\mathbf{x}, \mathbf{y})}{S_x S_y},$$

the corresponding projective geometric structure is as summarized in Table 1. Vector representations for $r_{pb}$ and $r_{pbd}$ are also listed, and a geometric visualization for $r_{pb}$ is shown in Fig 3. Consequently, $r_{pbd}$, Cohen's $d$, and $\rho_{pb}$ each possess $\mathbb{P}^1$ and $\mathbb{S}^1_+$ representations and serve as measures of group overlap, as described in section 2.2. Therefore, we conclude that the general classification of effect size as a 'relationship', 'group difference', or 'group overlap' index is misleading. We also observe that the question of the merits of Cohen's $d$ versus $r_{pb}$ in [3] is complicated by the fact that these measures correspond to points in different spaces, $\mathbb{P}^1$ and $\mathbb{S}^1_+$,

**Table 1. Homogeneous coordinates for Pearson correlation.**

| Effect size | $\mathbb{R}^2$ | $\mathbb{S}^1_+$ | $\mathbb{P}^1$ |
|---|---|---|---|
| Pearson correlation | $\left(\text{Cov}(\mathbf{x}, \mathbf{y}), \sqrt{S_{\mathbf{x}}^2 S_{\mathbf{y}}^2 - \text{Cov}^2(\mathbf{x}, \mathbf{y})}\right)t$ | $r = \dfrac{\text{Cov}(\mathbf{x}, \mathbf{y})}{S_{\mathbf{x}} S_{\mathbf{y}}}$ | $\dfrac{r}{\sqrt{1 - r^2}}$ |
| Point-biserial correlation | $\left(\sqrt{p_A p_B}(\bar{y}_A - \bar{y}_B), S_p\right)t$ | $r_{\text{pb}} = \dfrac{\sqrt{p_A p_B}d}{\sqrt{p_A p_B d^2 + 1}}$ | $\sqrt{p_A p_B}d$ |
| $r_{\text{pbd}}$ | $(\bar{y}_A - \bar{y}_B, 2S_p)t$ | $r_{\text{pbd}} = \dfrac{d}{\sqrt{d^2 + 4}}$ | $\dfrac{d}{2}$ |

The representations for Pearson product-moment correlation as homogeneous coordinates in $\mathbb{R}^2$, the vector $(r, \sqrt{1 - r^2}) \in \mathbb{S}^1_+$, and the vector $(r/\sqrt{1 - r^2}, 1) \in \mathbb{P}^1$. Corresponding representations for the point-biserial correlation, $r_{\text{pb}}$, and sample size adjusted correlation, $r_{\text{pbd}}$, are also listed. Cohen's $d = (\bar{y}_A - \bar{y}_B)/S_p$, $\{\bar{y}_A, \bar{y}_B\}$: mean values, $S_p$: pooled variance, $\{p_A, p_B\}$: sample size proportions for 'A' and 'B' data, $t \in \mathbb{R}^1$.

respectively. The limitations of $r_{\text{pb}}$ are more easily understood by considering its representation as the vector, $\left(\sqrt{p_A p_B}d, 1\right) \in \mathbb{P}^1$. The binomial factor has a confounding effect, particularly since base rates are determined by the experimental protocol. This is analogous to the confounding effect of the marginal sums on the $\phi$ coefficient for a $2 \times 2$ contingency table (Paper1). Therefore, neither $r_{\text{pb}}$ nor $\phi$ meet the criterion for a well-behaved effect size of serving to quantify 'some phenomenon that addresses a question of interest' [6]. In section 2.5, we give an example where $r_{\text{pb}}$ gives nonintuitive results in rCART analysis.

## 2.5 Point-biserial variation in regression tree analysis

The CART association graph was introduced in Paper1 as a new method for analyzing statistical association in point-biserial data. In this section, we investigate the role of point-biserial variation in rCART, particularly the connection between $\text{IG}_{\text{MSE}}$ and $r_{\text{pb}}$, and introduce the rCART graph as a new method for analyzing association for $(\mathbf{x}, \mathbf{y})$ data. The CART decision tree algorithm creates a decision tree by recursive partitioning of the association between response and independent variables [2, 14]. Each node of the tree corresponds to a binary partition of the range of an independent variable. In standard implementations, the partition

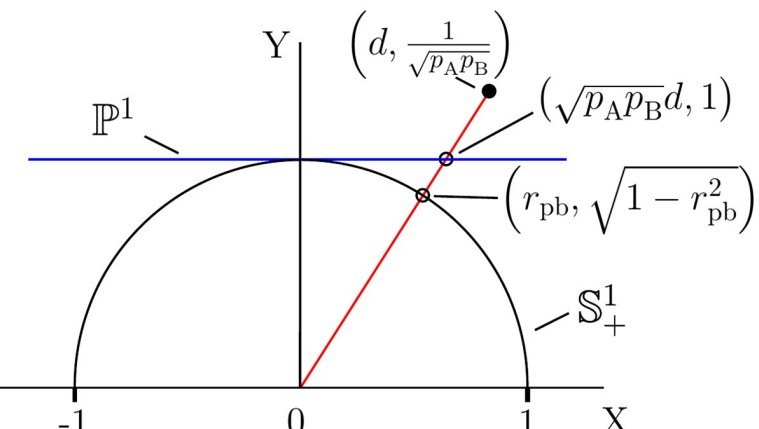

**Fig 3. Projective spaces for the representation of point-biserial correlation.** The point-biserial correlation coefficient, $r_{\text{pb}}$, corresponds to the point $(r_{\text{pb}}, \sqrt{1 - r_{\text{pb}}^2})$ on the positive half-circle, $\mathbb{S}^1_+$, and the point $(\sqrt{p_A p_B}d, 1)$ on the projective line, $\mathbb{P}^1$. The homogeneous coordinates $(\sqrt{p_A p_B}d, 1)t$ for $t \in \mathbb{R}^1$ correspond to points on the line through the origin. $\{p_A, p_B\}$: sample size proportions, $d$: Cohen's $d$.

parameters for a node are determined by maximizing the information gain (IG) for the response variable in an exhaustive search of associations over all independent variables. The rCART implementation is of particular interest because it involves the analysis of point-biserial variation. In each iteration, the set of statistics obtained for partitions of an independent variable constitutes a CART association graph [2]. For the partition value $x_j \in \mathbb{R}^1$, the data for a node ($V$) are divided into two subsets, i.e., $V_A = \{(x_i, y_i)|x_i \leq x_j\}$ and $V_B = \{(x_i, y_i)|x_i > x_j\}$, from which data vectors $\{\mathbf{y}_A, \mathbf{y}_B\}$ are obtained. Alternatively, if $x_j$ is categorical, the subsets are specified using matching criteria $V_A = \{(x_i, y_i)|x_i = x_j\}$ and $V_B = \{(x_i, y_i)|x_i \neq x_j\}$. The standard rCART impurity measure is the mean square error for the response, $\mathrm{MSE}(\mathbf{y}) = \sum_i(\bar{y} - y_i)^2/N_V$, where $N_V$ is the sample size and $\bar{y}$ is the mean [14]. Then, IG is defined as the parent node impurity minus the weighted impurities for the subsets

$$\mathrm{IG}_{\mathrm{MSE}}(\mathbf{y}_A, \mathbf{y}_B) = \mathrm{MSE}(\mathbf{y}) - \sum_{k=A,B} p_k \mathrm{MSE}(\mathbf{y}_k), \tag{19}$$

where $p_A$ and $p_B$ are the sample size proportions. Partitioning the sum of squares, $\mathrm{MSE}(\mathbf{y})$, gives [3, 21]

$$\begin{aligned} \mathrm{MSE}(\mathbf{y}) \quad &= \sum_{k=A,B} p_k(\bar{y}_k - \bar{y})^2 + \sum_{k=A,B} p_k \mathrm{MSE}(\mathbf{y}_k), \\ &= p_A p_B(\bar{y}_A - \bar{y}_B)^2 + \sum_{k=A,B} p_k \mathrm{MSE}(\mathbf{y}_k). \end{aligned}$$

Substitution for $\mathrm{MSE}(\mathbf{y})$ in Eq 19 gives

$$\mathrm{IG}_{\mathrm{MSE}}(\mathbf{y}_A, \mathbf{y}_B) = p_A p_B(\bar{y}_A - \bar{y}_B)^2. \tag{20}$$

Thus, $\mathrm{IG}_{\mathrm{MSE}}(\mathbf{y}_A, \mathbf{y}_B)$ is equivalent to $r_{\mathrm{pb}}^2$ with $S_p = 1$ (Table 1); $\mathrm{IG}_{\mathrm{MSE}}$ does not account for the variation in $S_p$. To the best of our knowledge, this connection between $\mathrm{IG}_{\mathrm{MSE}}$ and $r_{\mathrm{pb}}$ has not been reported previously. We conclude that the analysis of point-biserial variation serves as the basis for rCART, and we use the terms 'effect size' and 'information gain' interchangeably. The $x_j$ partition produces subsets with sample sizes, $j$ and $N_V - j$ for $x_j \in \mathbb{R}^1$. An association graph is obtained by searching over all partitions where the sample size proportions, $p_j$ and $(1 - p_j)$, vary over their entire range, producing a large parabolic variation in the $p_j(1 - p_j)$ factor. Thus, an association graph is a convenient way to compare the sample size proportion dependence of effect size measures. In the next section, we demonstrate that $r_{\mathrm{pb}}$ gives misleading results in rCART, while $r_{\mathrm{pbd}}$ and $\rho_{\mathrm{pb}}$ produce more intuitive results. However, when the $(\mathbf{x}, \mathbf{y})$ data are highly correlated and Pearson $r(\mathbf{x}, \mathbf{y}) \to 1$, the rCART graph becomes a horizontal line or nearly so, because $r_{\mathrm{pbd}} \approx \rho_{\mathrm{pb}} \approx 1$ for all $x_j$ partitions. Then, the rCART graph and Pearson $r$ are equivalent representations. Thus, CART methodology is most useful when the data are poorly correlated, which includes population studies where system performance is determined by trade-offs between multiple factors. Typical applications include GWAS, and other high-dimensional search problems such as nursing home performance as discussed in the next section.

## 3 Data analysis and results

In Paper1, we used the publicly accessible Nursing Home Compare (NHC) data [25] in CART analysis to demonstrate the importance of adjusting for the dependence on marginal sums for $2 \times 2$ contingency tables [2]. In this section, we use a similar NHC data set for a discussion of point-biserial variation and the rCART algorithm. Our objective is to provide a practical

demonstration of the limitations of $r_{pb}$ due to the confounding effect of unbalanced sample sizes and to compare the behaviors of $r_{pbd}$ and $\rho_{pb}$. We also discuss the importance of accounting for three degrees of freedom, $(r_{pbd}, \mu, \rho_{pb})$, and the use of Monte Carlo methods to estimate the joint distribution of statistical parameters.

### 3.1 rCART association graphs for NHC quality measures

NHC data of the fourth quarter of 2018 were retrieved for 20 quality measures ($Q_i$) for 15341 nursing homes; detailed descriptions of these continuous variables can be found on the NHC website [26]. A histogram of the nursing home occupancy is shown in Fig 4A. Since performance estimates for nursing homes with low occupancy would be less reliable, a minimum occupancy criterion of at least 50 'Average number of residents per day' was applied to obtain a restricted data set of 11053 nursing homes for further analysis [27]. Pearson correlation coefficients, $r(Q_i, Q_j)$, and association graphs were calculated for all pairs of quality measures, $\{(Q_i, Q_j)|i \neq j\}$. On average, the information gain for the rCART partition is larger when the $(Q_i, Q_j)$ variables are highly correlated (Fig 5A); the $r(Q_i, Q_j)$ correlations are distributed with 95% less than 0.16 and a maximum of 0.65. The distribution for 'Number of outpatient emergency department visits per 1000 long-stay resident days' ('Emergency visits') versus 'Number of hospitalizations per 1000 long-stay resident days' ('Hospitalizations') with correlation $r = 0.37$ is skewed, with a long tail towards larger values (Fig 4B). rCART association graphs are shown for the 'Hospitalizations' response and 'Emergency visits' partition variables (Fig 6A and 6B), and for the reverse, i.e., 'Emergency visits' response and 'Hospitalizations' partition variables (Fig 6C and 6D). The high correlation between $r_{pb}$ and $\sqrt{p_A p_B}$ ($r = 0.99$) is typical and indicates that variation in the binomial sampling factor overrides the smaller variation in Cohen's $d$ (Eq 2). We also note that the graphs for $r_{pb}$ and $IG_{MSE}$ (not shown) are superimposable, as

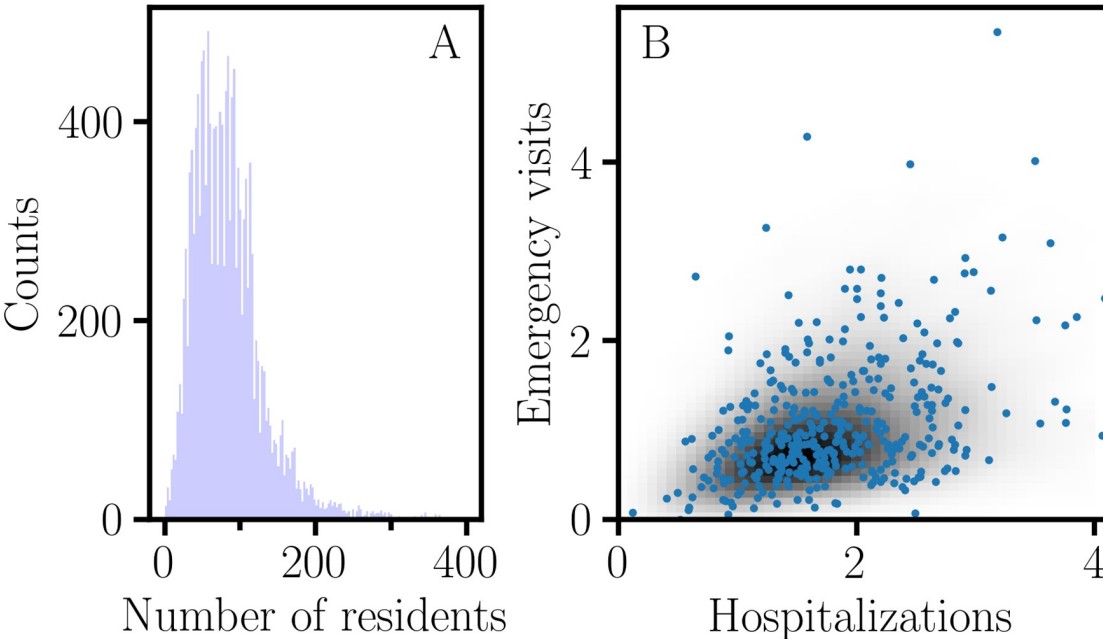

**Fig 4. Skewed distributions for NHC quality measures.** A. Histogram of 'Average number of residents per day' for 15341 nursing homes. B. Two-dimensional Gaussian kernel density estimate of the distribution of 'Number of outpatient emergency department visits per 1000 long-stay resident days' ('Emergency visits') versus 'Number of hospitalizations per 1000 long-stay resident days' ('Hospitalizations'), with correlation $r = 0.37$.

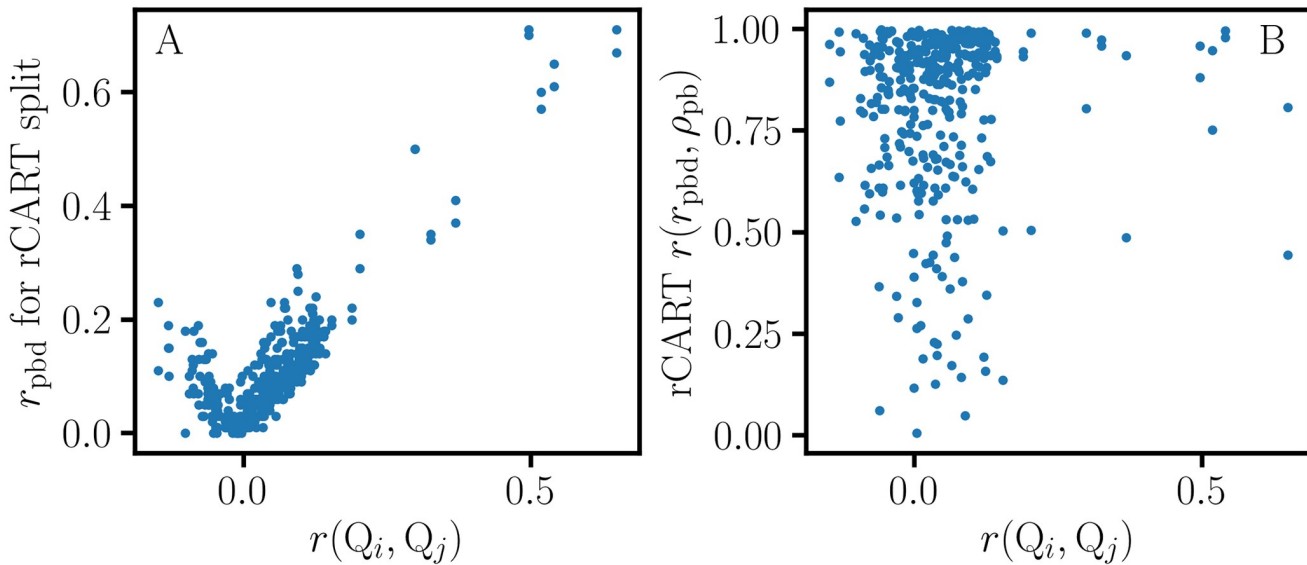

**Fig 5. The relation between $r_{pbd}$ and $\rho_{pb}$ in rCART.** These graphs display data obtained from association graphs for 380 pairs of quality measures, $\{(Q_i, Q_j)|i \neq j\}$. A. $r_{pbd}$ effect size for rCART split versus correlation $r(Q_i, Q_j)$. On average, the largest information gain is obtained when the response and partition variables are highly correlated. B. Correlation $r(r_{pbd}, \rho_{pb})$ between effect size and $r(Q_i, Q_j)$ for association graphs. There is good correlation between $r_{pbd}$ and $\rho_{pb}$ in many cases, but there are exceptions.

expected from Eq 20 and because the variation in $S_p$ is small. Thus, $r_{pb}$ and $IG_{MSE}$ mainly correspond to the variation in sample size proportion. In general, we observe that the association curves for $r_{pbd}$ and $\rho_{pb}$ can be categorized as monotonically increasing or decreasing, or even U-shaped (concave up), depending on how the $(Q_i, Q_j)$ data are distributed. Here, the U-shaped dependence of $r_{pbd}$ correlates well with $\delta$ ($r = 0.999$) and contrasts sharply with the concave down variation for $r_{pb}$. Consequently, $r_{pb}$ and $r_{pbd}$ produce very different rCART partitions (Table 2). In Fig 6A, the $r_{pb}$ partition for the split value, $x_j = 0.8$, produces subnodes with comparable sample sizes, $N_A = 5742$ and $N_B = 4890$ (Table 2). It is useful to view this partition from a statistical perspective. As a first approximation, we expect that the majority of nursing homes belong to a broad distribution for average performance. Then, the $r_{pb}$ partition with a split value close to the median, 0.85, is analogous to splitting a normal distribution nearly in half, producing subsets with different mean 'Emergency visits' values {0.5, 1.4} that nevertheless correspond to entities with average performance. Thus, $r_{pb}$ and $IG_{MSE}$ produce rCART subsets that are not well distinguished from a functional perspective. In comparison, for $r_{pbd}$, there are two possible rCART partitions at either low ($x_j = 0.3$) or high ($x_j = 2.5$) split values. Each partition produces a large subset corresponding to a broad distribution for average performance and a much smaller subset for either above- or below-average performance. Thus, $r_{pbd}$ produces more functionally relevant classifications.

The importance of accounting for variation in both degrees of freedom, $(r_{pbd}, \mu)$, is illustrated in Fig 6B and 6D. Here, $\mu$ is monotonically increasing, and one of the $r_{pbd}$ partitions might be preferred depending on $\mu$. However, this requires an assessment of the cost-benefit trade-offs for $(r_{pbd}, \mu)$ variation, which will depend on the particular application. A close correspondence between $r_{pbd}$ and $\rho_{pb}$ is observed in many cases, with $r(r_{pbd}, \rho_{pb}) \geq 0.8$ in 68% of the association graphs (Fig 5B), but there are many cases where they differ depending on how the $(Q_i, Q_j)$ data are skewed. Fig 6C shows an example of the difference between the $\rho_{pb}$ and $r_{pbd}$ curves with $r(r_{pbd}, \rho_{pb}) = 0.49$. The $r_{pbd}$ partition for the lower split value might be preferred because it is associated with higher $\rho_{pb}$, depending on how the cost-benefit trade-off is

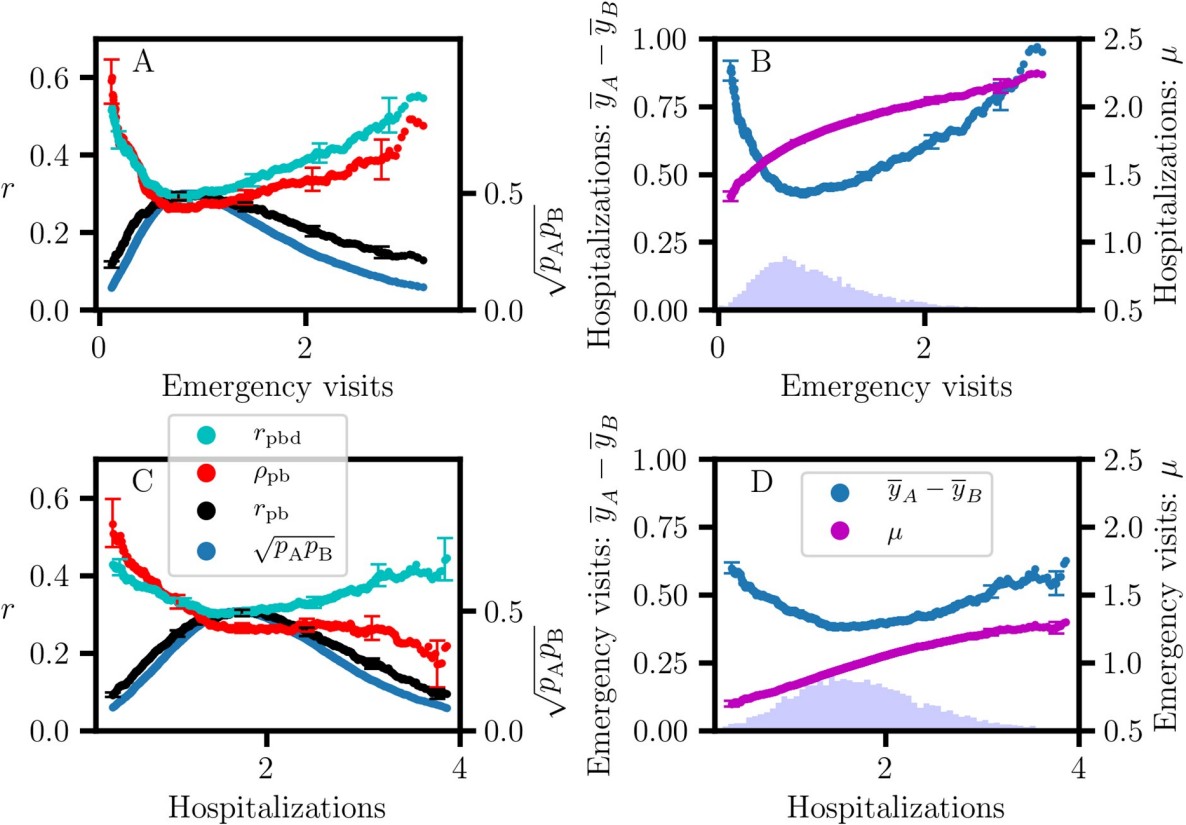

**Fig 6. rCART association graphs for effect size.** A,B: 'Hospitalizations' response versus 'Emergency visits' partition variables, with correlation $r(r_{\text{pbd}}, \rho_{\text{pb}}) = 0.93$. C,D: 'Emergency visits' response versus 'Hospitalizations' partition variables, with correlation $r(r_{\text{pbd}}, \rho_{\text{pb}}) = 0.49$. Bar plot histograms are shown for 'Emergency visits' (B inset) and 'Hospitalizations' (D inset). $r_{\text{pb}}$: point-biserial correlation coefficient, $\{p_A, p_B\}$: sample size proportions, $r_{\text{pbd}}$: sample size corrected correlation coefficient, $\rho_{\text{pb}}$: nonoverlap proportion, $(\delta, \mu)$: center of mass parameters $(\bar{y}_A - \bar{y}_B, (\bar{y}_A + \bar{y}_B)/2)$.

assessed for $(r_{\text{pbd}}, \rho_{\text{pb}})$ variation. Consequently, three coordinates $(r_{\text{pbd}}, \mu, \rho_{\text{pb}})$ are needed to distinguish different forms of point-biserial variation. These observations provide support for previous remarks stating that interpreting the magnitude of an effect size as a measure of substantive significance depends on the particular application [6, 18]. A more precise approach would take into account the multidimensional nature of point-biserial variation and involve the specification of functional or engineering requirements for a relevant vector basis. Then,

**Table 2. rCART subnode parameters.**

| Response variable | Partition variable | Split value | Subnode A | Subnode B |
|---|---|---|---|---|
| Hospitalizations | Emergency visits | $r_{\text{pbd}}$: 0.3 | 1.8, 0.7, 9909 | 1.2, 0.7, 723 |
| " | " | $r_{\text{pb}}$: 0.8 | 2.0, 0.7, 5742 | 1.5, 0.7, 4890 |
| " | " | $r_{\text{pbd}}$: 2.5 | 2.5, 0.8, 320 | 1.7, 0.7, 10312 |
| Emergency visits | Hospitalizations | $r_{\text{pbd}}$: 0.7 | 1.0, 0.6, 10137 | 0.5, 0.4, 495 |
| " | " | $r_{\text{pb}}$: 1.7 | 1.2, 0.7, 5318 | 0.8, 0.5, 5314 |
| " | " | $r_{\text{pbd}}$: 3.3 | 1.5, 1.0, 330 | 1.0, 0.6, 10302 |

Summary of the rCART partition values and subnode statistics $(\bar{y}, \sigma, N)$ for the association graphs in Fig 6. $r_{\text{pb}}$: point-biserial correlation coefficient, $r_{\text{pbd}}$: sample size corrected correlation, $(\bar{y}, \sigma, N)$: mean value, standard deviation, sample size.

an analysis of the effect size for the system response could involve separate thresholds for each coordinate. The ability to account for all relevant degrees of freedom is also important in assessing reproducibility. A one-parameter representation using an effect size such as $r_{pbd}$ or Cohen's $d$ gives an incomplete picture and leads to ambiguous results because of the loss of information.

## 3.2 Distributed effects in point-biserial variation

The reproducibility of nursing home performance data depends on stochastic effects in the measurement of patient outcome. Then, the observed data are associated with a distribution of data sets, $\mathcal{P}(\mathbf{y})$, and corresponding distributions of the statistical parameters $\mathcal{P}(\mathbf{v}_{pb})$ and effect size. The specification of $\mathcal{P}(\mathbf{y})$ must be based on a realistic assessment of all sources of error and uncertainty to form an error model for the data, $\mathcal{E}(\mathbf{y})$. Then, the determination of the distribution for the effect size requires propagation of the error in $\mathcal{P}(\mathbf{y})$. For fractional quantities such as Cohen's $d$ and $r_{pbd}$, it is necessary to account for stochastic effects in both the numerator and denominator. However, analytical methods for estimating distributions for ratios [28, 29], proportions [30, 31], and correlation coefficients [32] are complicated by fractional transformation, a bounded range, and discreteness. Thus, iterative procedures are needed for the analysis of noncentral effect size distributions and estimating confidence intervals for deviations above and below the effect size estimate [5, 18]. Alternatively, Monte Carlo (MC) methods [2, 33, 34] provide a more practical approach to estimating the distribution for the effect size. In an MC simulation, $\mathcal{E}(\mathbf{y})$ specifies error parameters for each observed value in the original data. Then, a point-biserial MC data set is obtained by random sampling to produce MC instances for $\mathbf{y}_A$ and $\mathbf{y}_B$. The MC sampling process is repeated many times to obtain a collection of MC data sets to form an estimate, $\mathcal{P}_i(\mathbf{y})$. Statistical parameters are calculated for the data sets in $\mathcal{P}_i(\mathbf{y})$ to obtain estimates of distributions and histograms for point-biserial effects. Many MC runs are performed to obtain a set, $\{\mathcal{P}_i(\mathbf{y})|1 \leq i \leq N_{MCruns}\}$, which allows the determination of the degree of convergence for the MC simulation. However, the information needed to construct an error model is not included in the NHC quality measures data. For this demonstration, we provided a rudimentary 'Emergency visits' error model, where $\sigma_i = y_i/5$. MC simulations for $(r_{pbd}, \mu)$ and $(r_{pbd}, \rho_{pb})$ for 'Emergency visits' response with 'Hospitalizations' rCART split value, 3.3 (Table 2), are shown in Fig 7. The discrete structure of the $\rho_{pb}$ distribution is due to stochastic effects in the $\mathbf{c_y}$ sorting. The separate confidence intervals in Fig 6 for positive and negative deviation from the observed effect size estimate were estimated from the MC distributions. In practical applications, the advantage of the MC method is that it allows detailed simulation of the data acquisition process, including heterogeneity within groups, and specifications for $\mathcal{E}(\mathbf{y})$ can include heteroscedasticity, measurement error, and misclassification [17, 35, 36].

## 4 Discussion

In this work, we use sort as an intrinsic property of both numbers and labels to generate a complete set of parameters for point-biserial variation, $\mathbf{v}_{pb}$. We demonstrate that Cohen's $d$ is associated with the center of mass representation for a two-component system of normal distributions. However, a parameterization can also be constructed for skewed distributions. We do not attempt to incorporate requirements for 'substantive significance' because this depends on the particular application, which might require different or additional parameters. The specification of performance criteria for all of the parameters in $\mathbf{v}_{pb}$ is also required. The $(\delta, \mu)$ effect size representation does not generalize because there is no standard center of mass parameterization for a multicomponent system. However, this does not constitute a

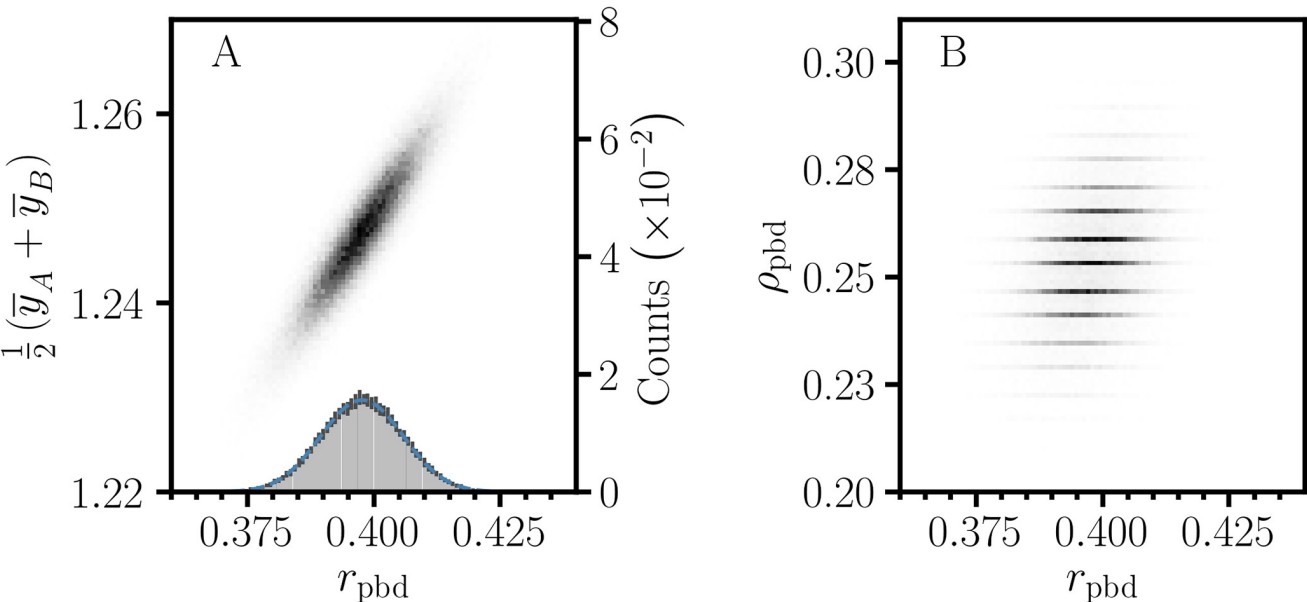

**Fig 7. Monte Carlo simulation of the distribution of stochastic effects for point-biserial variation.** 2D histograms of MC distributions for $(r_{pbd}, \mu)$ (A) and $(r_{pbd}, \rho_{pb})$ (B) for 'Emergency visits' response with 'Hospitalizations' rCART split value, 3.3 (Table 2). The $1\sigma$ error bars for the $r_{pbd}$ histogram (A inset) serve as an indication of convergence for the simulation; the mean for the normal curve corresponds to the observed $r_{pbd}$ value, 0.398. $r_{pbd}$: sample size corrected correlation, $\rho_{pb}$: nonoverlap proportion, $\mu$: center of mass parameter $(\bar{y}_A + \bar{y}_A)/2$, number of MC runs: 25, samples per MC run: 4000.

fundamental limitation in the application of effect size for high-dimensional data analytics. Instead, the $(\delta, \mu)$ coordinates serve as a minimal framework for analyzing dependency using exploratory methodologies such as rCART. CART methodology is useful in population studies where the performance or system response is distributed due to complex interactions. Then, a decision tree for identifying outperforming individuals can help in the determination of predictive criteria for improved performance, and the construction of a functional model. We also demonstrate the use of replication as a nonparametric method for equalizing sample sizes in the estimation of $\rho_{pb}$. This replication protocol can be used in other classification algorithms where adjustment for unbalanced sample size is needed. We also demonstrate that the Monte Carlo method is a practical way to estimate the distribution of a fractional statistical quantity from the detailed specification of an error model for the data. Then, the assessment of substantive significance must take into account the distribution in effect size parameters. We conclude that a better understanding of the applied algebraic foundations and an improved methodology are important for the application of effect size in data analytics.

## Acknowledgments

I thank many former colleagues in the Genetic Discovery group at DuPont for stimulating my interest in statistical problems in genome-wide association studies and CART.

## Author Contributions

**Conceptualization:** Stanley Luck.

**Formal analysis:** Stanley Luck.

**Investigation:** Stanley Luck.

**Methodology:** Stanley Luck.

**Software:** Stanley Luck.

**Visualization:** Stanley Luck.

**Writing – original draft:** Stanley Luck.

**Writing – review & editing:** Stanley Luck.

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
