## [Decision Letter · Decision Letter 0]

6 Oct 2020

PONE-D-20-22303

Nonoverlap proportion and point-biserial variation

PLOS ONE

Dear Dr. Luck,

Thank you for submitting your manuscript to PLOS ONE. After careful consideration, we feel that it has merit but does not fully meet PLOS ONE’s publication criteria as it currently stands. Therefore, we invite you to submit a revised version of the manuscript that addresses the points raised during the review process.

If major revisions are not met adequately this manuscript may be rejected upon resubmission. Please respond to the reviewer's comments on a point by point basis.

We look forward to receiving your revised manuscript.

Kind regards,

Alan D Hutson

Academic Editor

PLOS ONE

Journal Requirements:

Additional Editor Comments (if provided):

There is a substantial revision necessary.

Reviewers' comments:

Reviewer's Responses to Questions

**Comments to the Author**

1. Is the manuscript technically sound, and do the data support the conclusions?

Reviewer #1: Partly

Reviewer #2: Yes

2. Has the statistical analysis been performed appropriately and rigorously? 

Reviewer #1: No

Reviewer #2: Yes

3. Have the authors made all data underlying the findings in their manuscript fully available?

Reviewer #1: Yes

Reviewer #2: Yes

4. Is the manuscript presented in an intelligible fashion and written in standard English?

Reviewer #1: No

Reviewer #2: Yes

5. Review Comments to the Author

Reviewer #1: The paper especially the introductions and abstract are hard to read and follow which may due to the different established stats concepts mixed together, and these in general are applied to the different scenarios. The method section is too long, it’s unclear what the original work are, and what established techniques are discussed in the paper. This paper may be more suitable to submit the statistical types of journals based on the method section. The following are some detailed comments:

1. Title: is not informative, with only include two statistical terms.

2. Abstract: There are many notations in the abstract which have been messed up due to the formatting, and made the paper very hard to read and follow, e.g., what do R2, P1, etc. represent; why author choose to use rpb, ppb, rpbd types of abbreviation for the key terms of the paper. Readers will lose interests if contents re too technical in the abstract.

It should start with problem description or issues and challenges need be addressed, and then provide statistical solution with applications. The author may simplify the intro section and state clearly what the new work is proposed in this paper, and what are the associated applications.

3. Keywords, to many should limited to 5-6.

4. Introduction: page 2, author started with the Pearson product moment correlation coefficient (r) and δ (mean differences, the numerator of Cohen’s d), note that Pearson r and Cohen’s d are two well established stats concepts, each with specific formula to compute the effect size measures. They are applied for two different scenarios: the former is used to correlated two continuous variables (such as age and BMI); while the latter is used for one continuous and one binary (such as binary treatment (y/n) on blood pressure). It’s unclear what are the motivation to mix two scenarios and build the connection between these different effect size measures. It’s inappropriate to simply apply Pearson r in the case of one binary and one continuous variable situation which is why there are list of difficulties author mentioned from line 19-24. If the author intends to correlate the paired data such as before and after, he may need make it clear.

5. Line 16-17, p value answers hypothesis testing question of statistical significance, while effect size measure is related to estimation question, they are different stats techniques, and serves for different purpose, so they are not alternative. It’s true that p value has the drawback of sample size dependence, and the effect size shows the clinical/practical relevance/importance regardless of sample size.

6. Line 24-25:” dichotomy of a normal distribution”, do you mean bivariate normal?

7. Line 24 What does this means “In Paper1, “, is this published?

8. Line 24, “2 × 2 34 contingency tables” refer to two binary variables situations? Why is here/

9. It’s known taking the square the Pearson r value, one can obtain R^2, which shows the variations shared by 2 continuous variables, if R^2 subtracted by 1, one can get the no-overlap portion not shared by two variables, why your proposed novel “point-biserial variation” is needed?

10. Line 48, “paper1” mentioned second time, it appears without reading author’s “paper 1”, it’s difficult to connect and understand the techniques discussed here.

11. Method: it’s unclear what are the new/original work, and what are old/established techniques discussed here. Too long and hard to read.

12. It was poorly formatted, mixed with single spaced with double spaced. Figure’ title is separated from figure.

Reviewer #2: 1. There is no expansion for CART but the abbreviation is frequently used.

2. Proper citation to be used for paper 1

3. Keywords are missing

4.Prposed work explanation is not clear, So should explain clearly.

5. Need to check Spacing and Alignment of the paper

6.Add recent references.

6. PLOS authors have the option to publish the peer review history of their article (what does this mean?). If published, this will include your full peer review and any attached files.

Reviewer #1: No

Reviewer #2: **Yes: **Dr Kalpana Murugan, Professor & Head, Kalasalingam Academy of Research and Education, Krishnankoil, Virudhunagar(Dt) 626 126

---

## [Author Response · Author response to Decision Letter 0]

12 Nov 2020

The manuscript has been revised and the `Response to reviewers' document includes responses to each point raised by the reviewers.

Thanks.

---

## [Decision Letter · Decision Letter 1]

1 Dec 2020

PONE-D-20-22303R1

Nonoverlap proportion and the representation of point-biserial variation

PLOS ONE

Dear Dr. Luck,

Thank you for submitting your manuscript to PLOS ONE. After careful consideration, we feel that it has merit but does not fully meet PLOS ONE’s publication criteria as it currently stands. Therefore, we invite you to submit a revised version of the manuscript that addresses the points raised during the review process.

Please attend to some of the minor corrections. 

We look forward to receiving your revised manuscript.

Kind regards,

Alan D Hutson

Academic Editor

PLOS ONE

Reviewers' comments:

Reviewer's Responses to Questions

**Comments to the Author**

1. If the authors have adequately addressed your comments raised in a previous round of review and you feel that this manuscript is now acceptable for publication, you may indicate that here to bypass the “Comments to the Author” section, enter your conflict of interest statement in the “Confidential to Editor” section, and submit your "Accept" recommendation.

Reviewer #2: All comments have been addressed

2. Is the manuscript technically sound, and do the data support the conclusions?

Reviewer #2: Yes

3. Has the statistical analysis been performed appropriately and rigorously? 

Reviewer #2: Yes

4. Have the authors made all data underlying the findings in their manuscript fully available?

Reviewer #2: Yes

5. Is the manuscript presented in an intelligible fashion and written in standard English?

Reviewer #2: Yes

6. Review Comments to the Author

Reviewer #2: 1. In the revised manuscript the author mentioned the citation Paper1 [2]. Need clarification for this citation.

2. Remaining all the reviewer comments are incorporated .

7. PLOS authors have the option to publish the peer review history of their article (what does this mean?). If published, this will include your full peer review and any attached files.

Reviewer #2: **Yes: **Dr Kalpana Murugan

---

## [Author Response · Author response to Decision Letter 1]

9 Dec 2020

I would like to thank the reviewers again for their very helpful comments.

I have included remarks (line 14-19) that explain the connection between Paper1 and the current work, as requested by Dr. Murugan.

List of additional minor corrections

- line 33, correction in the citation

- line 39, correction

- line 95, correction

- line 288, correction

- typo, reference #1, in References

---

## [Editor Report · Decision Letter 2]

11 Dec 2020

Nonoverlap proportion and the representation of point-biserial variation

PONE-D-20-22303R2

Dear Dr. Luck,

We’re pleased to inform you that your manuscript has been judged scientifically suitable for publication and will be formally accepted for publication once it meets all outstanding technical requirements.

Kind regards,

Alan D Hutson

Academic Editor

PLOS ONE
---

## [Editor Report · Acceptance letter]

15 Dec 2020

PONE-D-20-22303R2 

Nonoverlap proportion and the representation of point-biserial variation 

Dear Dr. Luck:

I'm pleased to inform you that your manuscript has been deemed suitable for publication in PLOS ONE. Congratulations! Your manuscript is now with our production department. 

Kind regards, 

on behalf of

Dr. Alan D Hutson 

Academic Editor

PLOS ONE